 **eLIFE**

# Autophagy functions as an antiviral mechanism against geminiviruses in plants

Yakupjan Haxim[1†], Asigul Ismayil[1†], Qi Jia[1], Yan Wang[1], Xiyin Zheng[1], Tianyuan Chen[1], Lichao Qian[1], Na Liu[1], Yunjing Wang[1], Shaojie Han[1], Jiaxuan Cheng[1], Yijun Qi[1], Yiguo Hong[2], Yule Liu[1]*

[1]Center for Plant Biology, Tsinghua-Peking Joint Center for Life Sciences, MOE Key Laboratory of Bioinformatics, School of Life Sciences, Tsinghua University, Beijing, China; [2]Research Centre for Plant RNA Signaling, College of Life and Environmental Sciences, Hangzhou Normal University, Hangzhou, China

**Abstract** Autophagy is an evolutionarily conserved process that recycles damaged or unwanted cellular components, and has been linked to plant immunity. However, how autophagy contributes to plant immunity is unknown. Here we reported that the plant autophagic machinery targets the virulence factor $\beta$C1 of *Cotton leaf curl Multan virus* (CLCuMuV) for degradation through its interaction with the key autophagy protein ATG8. A V32A mutation in $\beta$C1 abolished its interaction with NbATG8f, and virus carrying $\beta$C1$^{V32A}$ showed increased symptoms and viral DNA accumulation in plants. Furthermore, silencing of autophagy-related genes *ATG5* and *ATG7* reduced plant resistance to the DNA viruses CLCuMuV, *Tomato yellow leaf curl virus*, and *Tomato yellow leaf curl China virus*, whereas activating autophagy by silencing *GAPC* genes enhanced plant resistance to viral infection. Thus, autophagy represents a novel anti-pathogenic mechanism that plays an important role in antiviral immunity in plants.

*For correspondence: yuleliu@ mail.tsinghua.edu.cn

[†]These authors contributed equally to this work

Competing interests: The authors declare that no competing interests exist.

## Introduction

Plants have evolved various defense mechanisms to combat plant pathogens, including viruses. The two major mechanisms for plant antiviral immunity are RNA silencing and resistance (*R*) gene-mediated resistance (*Mandadi and Scholthof, 2013*). RNA silencing is a sequence-specific mechanism used to directly defend host cells against foreign invaders such as viruses and transposable elements (*Ding, 2010*). By contrast, the activation of *R* gene-mediated resistance triggers a rapid defense response that often includes localized programmed cell death, known as the hypersensitive response (HR). The HR can prevent local viral infection and elicit systemic acquired resistance to viral infection.

Autophagy is an evolutionarily conserved mechanism that recycles damaged or unwanted cellular materials under stress conditions or during specific developmental processes (*Liu and Bassham, 2012*), and plays a critical role in multiple physiological processes, including plant biotic stress responses (*Han et al., 2011*). During the plant's response to incompatible pathogens, autophagy contributes to HR cell death but restricts the spread of programmed cell death beyond the initial infection site (*Liu et al., 2005*; *Patel and Dinesh-Kumar, 2008*; *Hofius et al., 2009*; *Yoshimoto et al., 2009*). During compatible plant–pathogen interactions, autophagy positively regulates plant defense responses against necrotrophic pathogens (*Lai et al., 2011*; *Lenz et al., 2011*; *Kabbage et al., 2013*). However, disrupting autophagy in *Arabidopsis thaliana* leads to enhanced resistance to the biotrophic pathogen powdery mildew and dramatic pathogen-induced cell death (*Wang et al., 2011*). The role of autophagy in plant defense responses against the bacterial pathogen *Pseudomonas syringae* DC3000 is controversial (*Patel and Dinesh-Kumar, 2008*; *Hofius et al., 2009*; *Lenz et al., 2011*). However, it is unclear how autophagy links plant immunity in these studies.

**eLife digest** Plants use a variety of processes to protect themselves against viruses and other disease-causing microbes. Autophagy, for example, is a process that breaks down damaged or unwanted molecules found inside cells, which has also been linked to plant disease resistance. However, it is not precisely clear how autophagy helps plants to resist diseases, because this process seems to make plants more resistant to some disease-causing microbes but more susceptible to others.

Now, Haxim, Ismayil et al. show that autophagy helps to protect plants against three viruses belonging to the *Geminiviridae* family of plant viruses. One of these viruses causes an important disease in cotton plants, called cotton leaf curl disease. This virus can infect many other plant species, including a close relative of tobacco plants, called *Nicotiana benthamiana,* which is commonly used in plant biology experiments. Haxim, Ismayil et al. show that one of proteins produced by this virus, one called βC1, interacts with a plant protein called ATG8 and is then sent to be broken down by autophagy. Further experiments then identified a mutation in this protein that stopped it interacting with ATG8. Viruses carrying this mutated form of βC1 caused more severe symptoms and replicated more in *N. benthamiana* plants.

Interfering with autophagy made the *N. benthamiana* plants less resistant to the cotton leaf curl disease virus, and to two other geminiviruses that often infect tomatoes. Activating autophagy had the opposite effect, and made the plants more resistant to all three viruses.

Together these findings provide direct evidence that autophagy helps to defend plants against a number of viruses, by degrading one or more viral proteins in the plants. In the future, researchers may be able to build on these findings to engineer crop plants to be more resistant to viruses.

Autophagy may link plant immunity in different ways, with autophagy playing a role in degrading pathogen effectors or defense-related plant proteins, or pathogen effectors interfering with autophagy. Indeed, viral proteins are reported to promote autophagic degradation of plant host RNAi-related components (*Derrien et al., 2012*; *Cheng and Wang, 2016*). In addition, 2b protein from *Cucumber mosaic virus* is thought to be targeted for degradation by autophagy through the calmodulin-like protein rgsCaM (*Nakahara et al., 2012*). Recently, an oomycete effector is reported to interfere with autophagy by depleting the putative selective autophagy cargo receptor Joka2 out of ATG8 complexes (*Dagdas et al., 2016*). However, the role of autophagy in degrading pathogen effectors or plant defense-related proteins and the effect of viral effectors on autophagy remain uncertain in plants. All current findings are based on the data from chemical autophagy inhibitor treatments (with potential off-target effects) and/or silencing of autophagy-nonspecific autophagy-related (ATG) gene *Beclin 1* (*Derrien et al., 2012*; *Nakahara et al., 2012*; *Cheng and Wang, 2016*). Further, in these above studies, no data showed that disruption of classic autophagy really affects pathogen invasion. Moreover, to date, there is no evidence to show that autophagy has a role during any compatible plant-virus interactions.

Geminiviruses are a large, diverse group of plant viruses with circular single-stranded DNA genomes, and many geminiviruses cause devastating diseases in different crops. These viruses often occur in disease complexes. For example, *Cotton leaf curl Multan virus* (CLCuMuV), in association with the disease-specific satellite DNA Cotton leaf curl Multan betasatellite (CLCuMuB), causes cotton leaf curl disease, a major viral disease in cotton (*Sattar et al., 2013*). In addition to cotton, CLCuMuV infects many other plants, including *Nicotiana benthamiana*. CLCuMuV encodes six proteins, namely C1, C2, C3, C4, V1 and V2, whereas CLCuMuB is approximately half the size of the CLCuMuV DNA genome and encodes a single protein, βC1 (*Briddon et al., 2003*). Like most βC1 factors encoded by geminivirus betasatellites, CLCuMuB βC1 is required by CLCuMuV for the induction of disease symptoms in plants. In addition, CLCuMuB βC1 enhances the accumulation of its helper virus, CLCuMuV (*Saeed et al., 2015*; *Jia et al., 2016*), and is involved in RNA silencing (*Amin et al., 2011*). Recently, we and other groups reported that geminivirus βC1s can subvert ubiquitination to assist their helper viruses to infect plants (*Jia et al., 2016*; *Shen et al., 2016*).

In this study, we demonstrated that autophagy targets the virulence factor $\beta$C1 of CLCuMuV for degradation. Furthermore, we uncovered that autophagy functions as a novel antiviral mechanism against three geminiviruses in plants.

## Results

### CLCuMuB $\beta$C1 interacts with the autophagy-related protein ATG8

To investigate the role of CLCuMuB $\beta$C1 (hereafter $\beta$C1) in plant–virus interactions, we performed yeast two-hybrid screening of a *Solanum lycopersicum* cDNA library using $\beta$C1 as the bait. From this screen, we identified the autophagy-related protein SlATG8f as a $\beta$C1-interacting protein. We also found that $\beta$C1 interacted with NbATG8f, the closest *N. benthamiana* homolog of SlATG8f in yeast (*Figure 1A*).

We validated the in vivo and in vitro interactions between $\beta$C1 and NbATG8f using Glutathione S-Transferase (GST) pull-down (*Figure 1B*) and co-immunoprecipitation assays (*Figure 1C*). More-over, we identified an 11-amino acid motif from residue 30 to 40 of $\beta$C1 that is necessary for its interaction with NbATG8f (*Figure 1—figure supplement 1*). Strikingly, GST pull-down and co-immu-noprecipitation assays indicated that the mutant protein $\beta$C1$^{V32A}$ was unable to interact with NbATG8f (*Figure 1B,C*). Further, co-immunoprecipitation assays indicated that $\beta$C1 also interacted with other three ATG8 isoforms (*Figure 1—figure supplement 2*).

We then performed a bimolecular fluorescence complementation (BiFC) assay to identify the sub-cellular localization of the $\beta$C1–NbATG8f interaction in plant cells. A positive interaction between nYFP-$\beta$C1 and cYFP-NbATG8f was observed in both the cytoplasm and vacuoles of plant cells, as indicated by the presence of yellow fluorescence (*Figure 1D*). However, no such interaction was detected between nYFP-$\beta$C1$^{V32A}$ and cYFP-NbATG8f (*Figure 1D*), although all constructs were suc-cessfully expressed (*Figure 1E*). We further confirmed the vacuolar localization of the $\beta$C1–NbATG8f interaction by performing time-lapse observations of mesophyll cells by confocal microscopy, which revealed Brownian motion of fluorescent punctate structures within the central vacuole (*Video 1*).

In addition, we found that YFP-$\beta$C1, but not YFP-$\beta$C1$^{V32A}$, co-localized with GFP-NbATG8f-posi-tive bodies in both the cytoplasm and central vacuoles of mesophyll cells of *N. benthamiana* leaf tis-sue, as revealed by confocal microscopy (*Figure 1—figure supplement 3*). The vacuolar deposition of YFP-$\beta$C1 was also confirmed by time-lapse observations of mesophyll cells by confocal micros-copy, which revealed Brownian motion of fluorescent punctate structures within the central vacuole (*Video 2*), which is consistent with the vacuolar localization of the $\beta$C1–NbATG8f interaction (*Video 1*).

Taken together, these results demonstrate that $\beta$C1 specifically interacts with NbATG8s and the V32 residue is essential for the $\beta$C1–NbATG8f interaction.

### Autophagy is induced during CLCuMuV infection

Since $\beta$C1 specifically interacted with NbATG8s, we hypothesized that autophagy affects the infec-tion of plants by geminiviruses. To test this hypothesis, we investigated whether geminivirus infec-tion could induce autophagy in *N. benthamiana*. First, we performed quantitative RT-PCR (qRT-PCR) analysis, finding that mRNA levels of *NbATG2*, *NbATG3*, *NbATG5*, and *NbATG7* were upregulated during the infection of plants with CLCuMuV (CA) plus CLCuMuB ($\beta$) (hereafter referred to as CLCu-MuV infection or CA+$\beta$) (*Figure 2—figure supplement 1*). We then used Cyan Fluorescent Protein (CFP)-tagged NbATG8f (CFP-NbATG8f) as an autophagosome marker (*Han et al., 2015*) to visualize possible autophagic activity. In *N. benthamiana* plants infected with CA+$\beta$, we observed increased numbers of autophagosomes (represented by CFP-NbATG8f puncta) (*Figure 2A,B*). Transmission electron microscopy confirmed that viral infection increased the number of autophagic structures in infected cells (*Figure 2C,D*) compared to the control. Further, we tested the effect of CLCuMuV infection on autophagy flux using Joka2/NBR1, a selective autophagy cargo receptor, as a protein marker. Joka2 has been used as a useful tool to measure autophagy flux in plants because it is degraded in the vacuole once autophagy activity increases (*Zhou et al., 2013*; *Xu et al., 2017*). Indeed, we found that CLCuMuV infection (CA+$\beta$) reduced the protein level of NbJoka2, although it did not change mRNA level of NbJoka2 (*Figure 2—figure supplement 2*), indicating that viral infec-tion enhances autophagic flux. These data demonstrate that CLCuMuV infection induces autophagy.

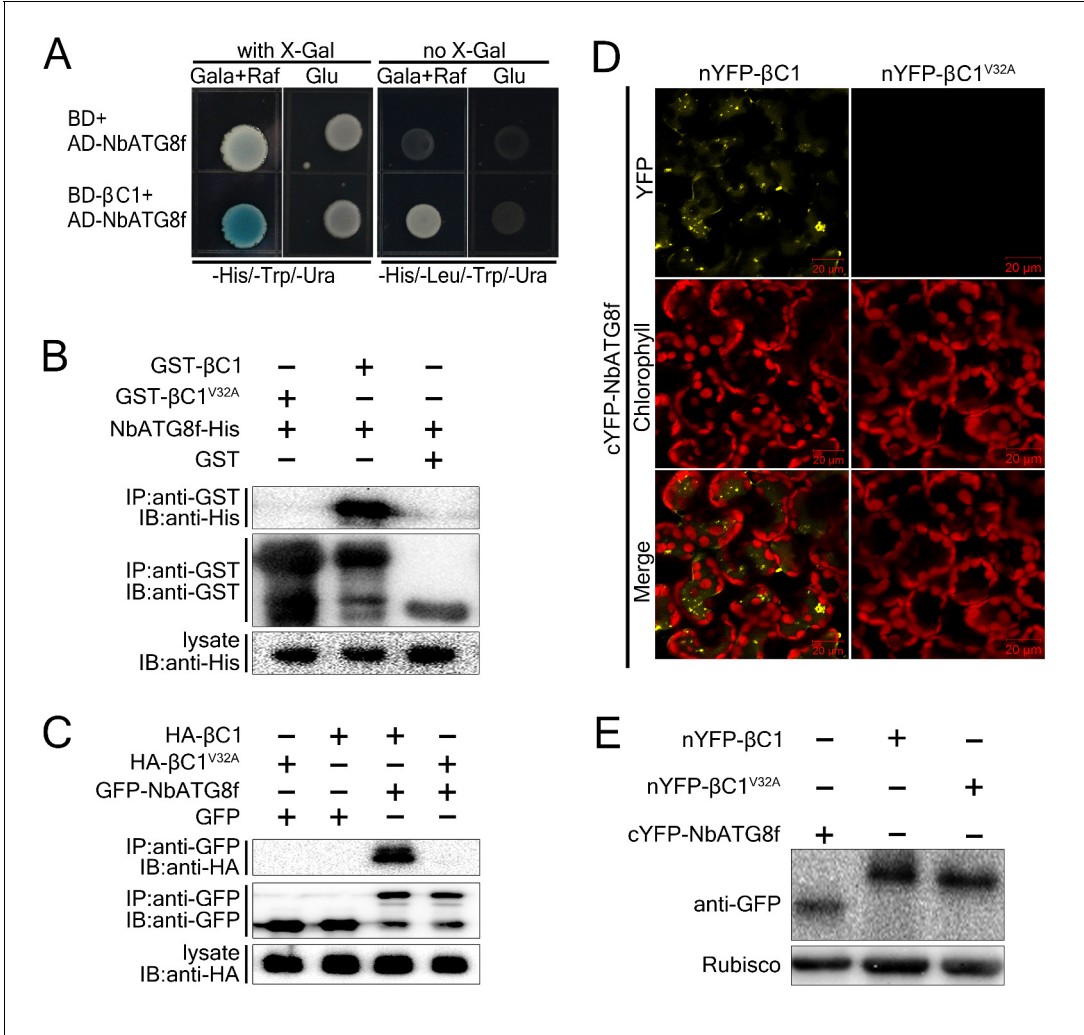

**Figure 1.** CLCuMuB *β*C1 interacts with NbATG8f in vivo and in vitro. (**A**) *β*C1 interacts with NbATG8f in yeast. SKY48 yeast strains containing AD-NbATG8f transformed with BD-*β*C1 or BD (control) were grown on Leu- selection plates at 28°C for 4 d. The positive interaction was indicated by the blue colony formation on X-gal-containing galactose (Gala) and raffinose (Raf) but not on plates containing glucose (Glu). (**B**) GST pull-down assay to show the in vitro interaction of NbATG8f with *β*C1, but not *β*C1$^{V32A}$. The total soluble proteins of *E. coli* expressing NbATG8f-6×His were incubated with GST-*β*C1 or GST-*β*C1$^{V32A}$ immobilized on glutathione-sepharose beads and monitored by anti-His antibody. (**C**) *β*C1 was co-immunoprecipitated with NbATG8f. GFP-NbATG8f was transiently co-expressed with and HA-*β*C1 or its mutant HA-*β*C1$^{V32A}$ in *N. benthamiana* leaves. At 60 hr post agroinfiltration (hpi), leaf lysates were immunoprecipitated with anti-GFP beads and then the precipitants were assessed by immunoblotting (IB) using anti-HA (upper panel) or anti-GFP antibodies (middle panel). (**D**) BiFC analyses in *N. benthamiana*. Representative images of nYFP-*β*C1 or nYFP-*β*C1$^{V32A}$ BiFC co-expressed with cYFP-NbATG8f. (**E**) Western blot analyses of BiFC construct combinations from the same experiments as in (**D**). All combinations were detected with anti-GFP polyclonal antibody.

The following figure supplements are available for figure 1:

**Figure supplement 1.** N terminus of *β*C1 is responsible for binding to NbATG8f.

**Figure supplement 2.** *β*C1 co-immunoprecipitated with multiple ATG8 homologs.

**Figure supplement 3.** *β*C1 is co-localized with NbATG8f.

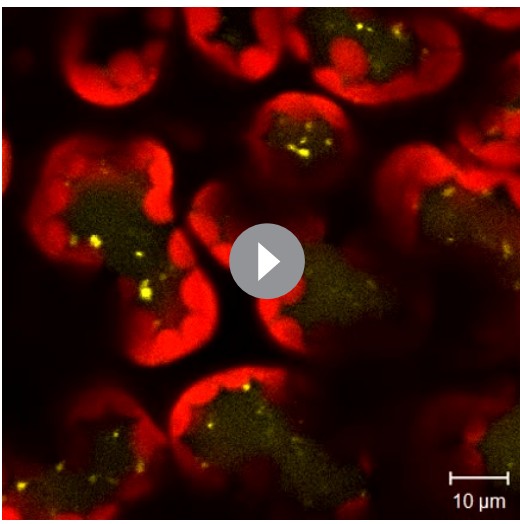

**Video 1.** βC1- NbATG8f Interaction Is Localized in Vacuoles. nYFP-βC1 transiently expressed with cYFP-NbATG8f in *N. benthamiana* leaves and examined by confocal laser scanning microscopy at 60 hpi. Yellow color represents YFP fusion fluorescence and red color for chlorophyll.

## Autophagy functions as an antiviral defense mechanism against CLCuMuV

Since autophagy is induced by CLCuMuV infection and βC1 specifically interacts with NbATG8f, we investigated the role of autophagy in CLCuMuV infection. For this purpose, we silenced autophagy-related genes in *N. benthamiana* using *Tobacco rattle virus* (TRV)-based virus-induced gene silencing (VIGS) (*Liu et al., 2002*). Because there is functional redundancy of *ATG8* genes due to the presence of multiple homologs in plants, we silenced *NbATG5* and *NbATG7* in *N. benthamiana*. Compared to non-silenced control plants, the mRNA levels of *NbATG5* and *NbATG7* were significantly reduced by gene-specific VIGS (*Figure 3—figure supplement 1A*), whereas we observed no obvious differences in TRV RNA levels between *ATG5*-silenced plants, *ATG7*-silenced plants, and control plants (*Figure 3—figure supplement 1B*). In addition, autophagy was blocked in *ATG5*- and *ATG7*-silenced plants (*Figure 3—figure supplement 2A,B*). It is worth noting that CLCuMuV infection had no effect on TRV-mediated VIGS (*Figure 3—figure supplement 3*). Furthermore, *ATG5*- and *ATG7*-silenced plants did not show any abnormal developmental phenotypes. We then infected *ATG5*- and *ATG7*-silenced plants with CA+β. We observed that the leaf curl symptoms caused by viral infection were much more severe and appeared 3 days earlier (*Figure 3A,B*), and CLCuMuV DNA levels were significantly higher in *ATG5*- and *ATG7*- silenced plants compared to control plants (*Figure 3C*). By contrast, silencing of *GFP* in a *N. benthamiana GFP* transgenic line 16C had no effect on CA+β infection (*Figure 3—figure supplement 4*).

Next, we examined the effect of enhanced autophagy on CLCuMuV infection. Consistent with the observation that down-regulating cytosolic glyceraldehyde-3-phosphate dehydrogenase (*GAPC*s) gene expression significantly activates autophagy (*Han et al., 2015*), we found that VIGS of *GAPC*s delayed symptom development in plants infected by CA+β (*Figure 3D,E*) and reduced viral DNA accumulation (*Figure 3F*). These results suggest that autophagy functions as an antiviral mechanism against CLCuMuV.

## The βC1-NbATG8f protein interaction is involved in antiviral defense against CLCuMuV infection

To further explore the biological significance of the βC1-NbATG8f interaction on autophagy-mediated defense against CLCuMuV infection, we generated a CLCuMuB mutant (β^V32A) by replacing βC1 with its mutant counterpart

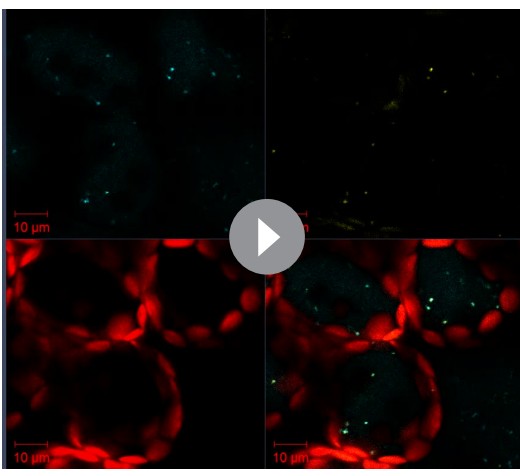

**Video 2.** βC1 Is Co-localized with NbATG8f. YFP-βC1 transiently expressed with CFP-NbATG8f in *N. benthamiana* leaves and examined by confocal laser scanning microscopy at 60 hpi. Cyan color represents CFP-ATG8f, yellow color YFP-βC1 and red color for chlorophyll.

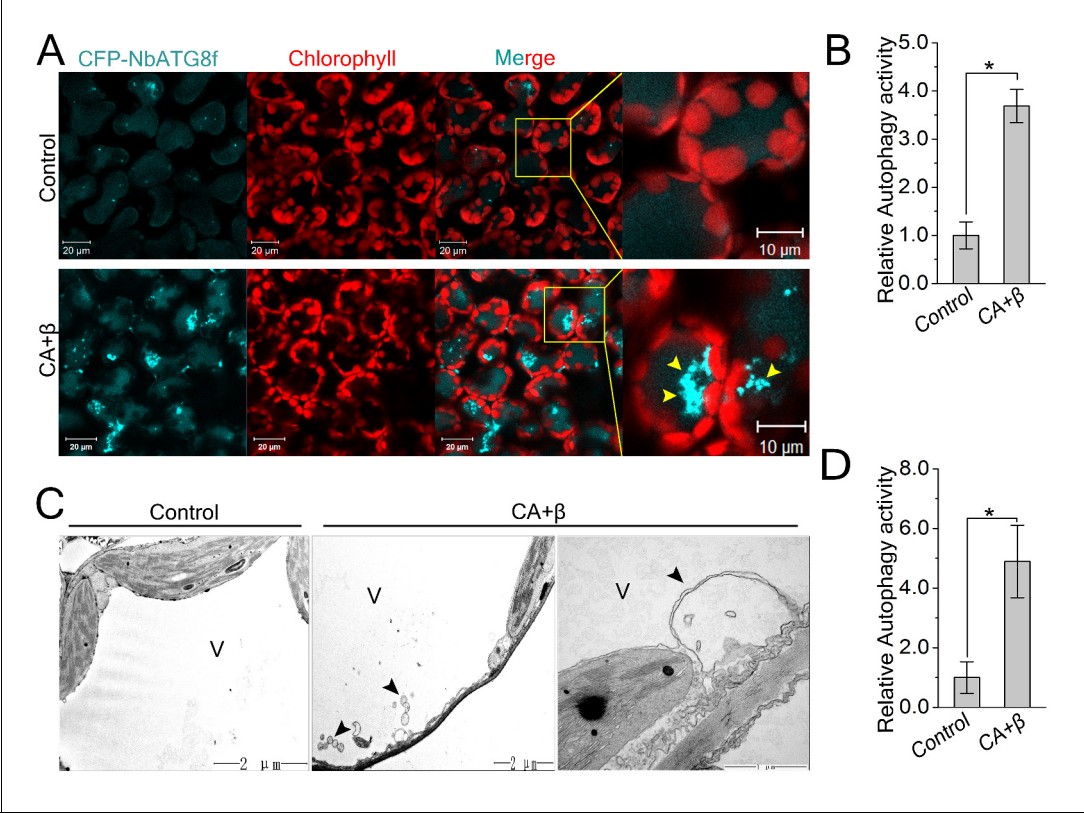

**Figure 2.** CLCuMuV infection activates autophagy. (**A**) Representative confocal microscopy images of dynamic autophagic activity revealed by specific autophagy marker CFP-NbATG8f in plants infected with CLCuMuV plus CLCuMuB (CA+$\beta$). (**B**) Quantification of the CFP-NbATG8f-labeled autophagic puncta per cell from (**A**). More than 500 mesophyll cells for each treatment were used for the quantification. Relative autophagic activity in virus infected plants was normalized to that of control plants, which was set to 1.0. Values represent means ± SE from three independent experiments. (*) p<0.05. (**C**) Representative TEM images of autophagic structures. Ultrastructure of autophagic bodies (arrows) was observed in the vacuoles of mesophyll cells of uninfected control and plants infected with CA+$\beta$. V for vacuole. (**D**) Autophagosome-like structures from (**C**) were quantified. At least 30 cells for each treatment were used for the quantification. Relative autophagic activity in virus infected plants was normalized to that of control plants, which was set to 1.0. Values represent means ± SE from three independent experiments. (*) p<0.05.

The following figure supplements are available for figure 2:

**Figure supplement 1.** Transcription pattern of autophagy-related genes were altered during CLCuMuV infection.

**Figure supplement 2.** Viral infection decreased NbJoka2/NBR1 protein level.

$\beta C1^{V32A}$ and inoculating this mutant virus (CA+$\beta^{V32A}$) onto *N. benthamiana* leaves. Interrupting the interaction between $\beta$C1 and ATG8 accelerated the occurrence of viral symptoms and resulted in much more severe leaf curling symptoms than that caused by CA+$\beta$ (*Figure 4A,B*). Leaf curling symptoms caused by CA+$\beta^{V32A}$ appeared 3 days earlier than the symptoms caused by CA+$\beta$ (*Figure 4A–B*). Moreover, we observed increased viral DNA accumulation in plants infected by CA +$\beta^{V32A}$ versus CA+$\beta$ (*Figure 4C*). Since the V32A point mutation eliminates the interaction of $\beta$C1 with NbATG8f, these results suggest that the interaction of $\beta$C1 with NbATG8 is essential for the antiviral defense mechanism of autophagy against CLCuMuV infection.

## $\beta$C1 is targeted by autophagy for its degradation

Since we observed the vacuolar localization of the $\beta$C1-NbATG8f interaction and co-localization of $\beta$C1 with NbATG8f, we guess that $\beta$C1 is delivered to the vacuoles by autophagy for its degradation. To test this hypothesis, we investigated the effect of autophagy on the subcellular localization of $\beta$C1 by expressing YFP-$\beta$C1 or its mutant YFP-$\beta$C1$^{V32A}$ in the non-silenced control, *ATG5*- and

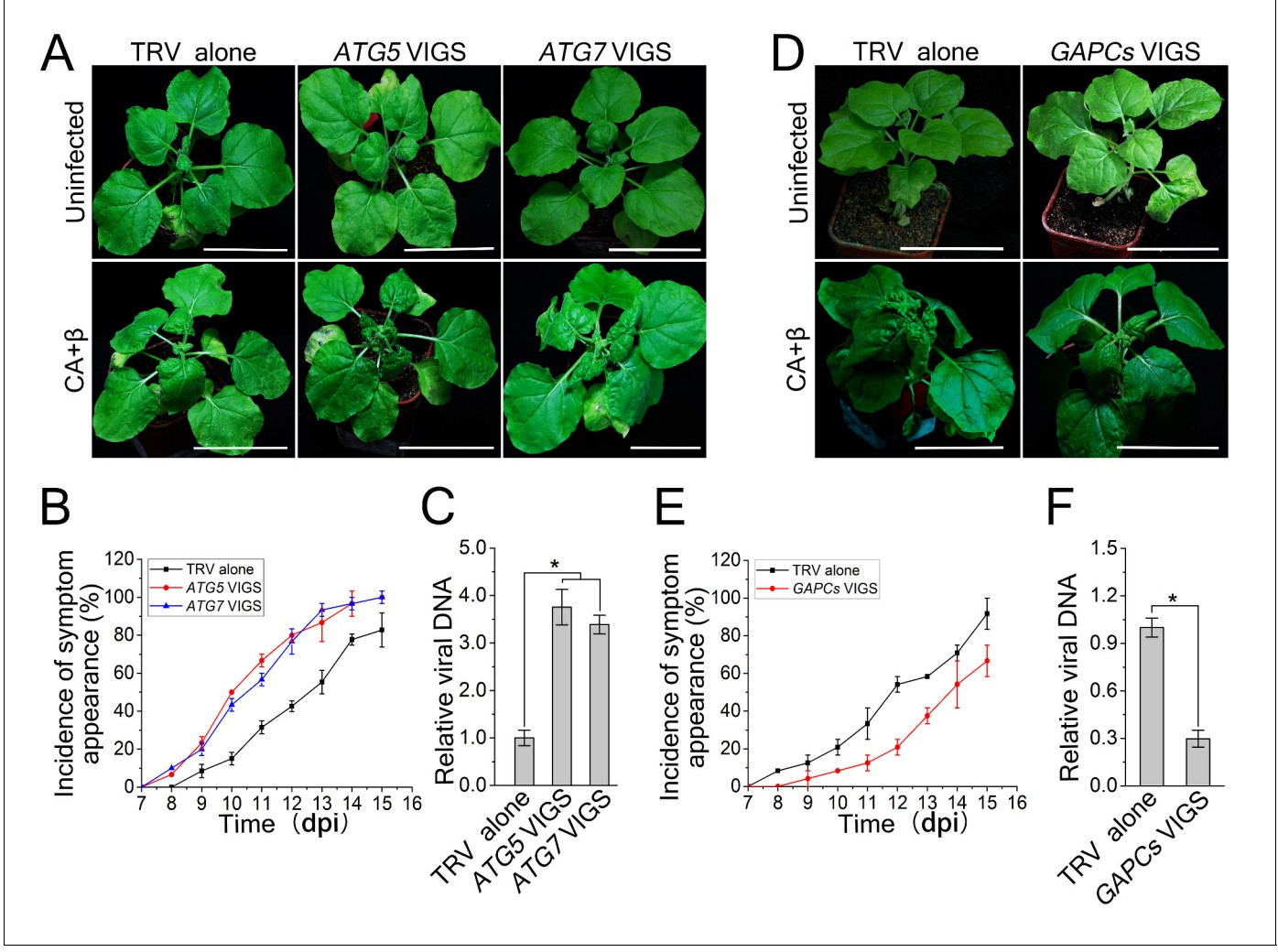

**Figure 3.** CLCuMuV DNA accumulation is affected by host cell autophagy. (**A**) Viral symptoms in *ATG5*- and *ATG7*–silenced plants infected with CLCuMuV plus CLCuMuB (CA+*β*) at 12 dpi. Bar represents 7 cm. (**B**) The incidence of viral symptom appearance at different time points of post infection in *ATG5*- and *ATG7*-silenced plants. Symptom was indicated as the appearance of curled leaf caused by CA+*β*. Values represent means ± SE from three independent experiments. (**C**) Relative viral DNA accumulation in *ATG5*- and *ATG7*–silenced plants infected with CA+*β*. Real-time PCR analysis of *V1* gene from CLCuMuV was used to determine viral DNA level. Values represent means ± SE from three independent experiments. (*) $p < 0.05$. (**D**) Viral symptoms in *GAPCs*–silenced plants infected with CA+*β* at 15 dpi. Bar represents 7 cm. (**E**) The incidence of symptom appearance at different time points of post infection in *GAPCs*-silenced plants. Symptom was indicated as the appearance of curled leaf caused by CLCuMuV infection. Values represent means ± SE from three independent experiments. (**F**) Relative viral DNA accumulation in *GAPCs*-silenced plants. Real-time PCR analysis of *V1* gene from CLCuMuV was used to determine viral DNA level. Values represent means ± SE from three independent experiments. (*) $p < 0.05$.

The following figure supplements are available for figure 3:

**Figure supplement 1.** TRV viral titers were not changed during VIGS.

**Figure supplement 2.** Involvement of *ATG5* and *ATG7* in autophagy is confirmed by VIGS.

**Figure supplement 3.** CLCuMuV infection has no effect on TRV-based VIGS of *NbPDS*.

**Figure supplement 4.** Silencing of a non-autophagy related gene *GFP* has no effect on CLCuMuV infection.

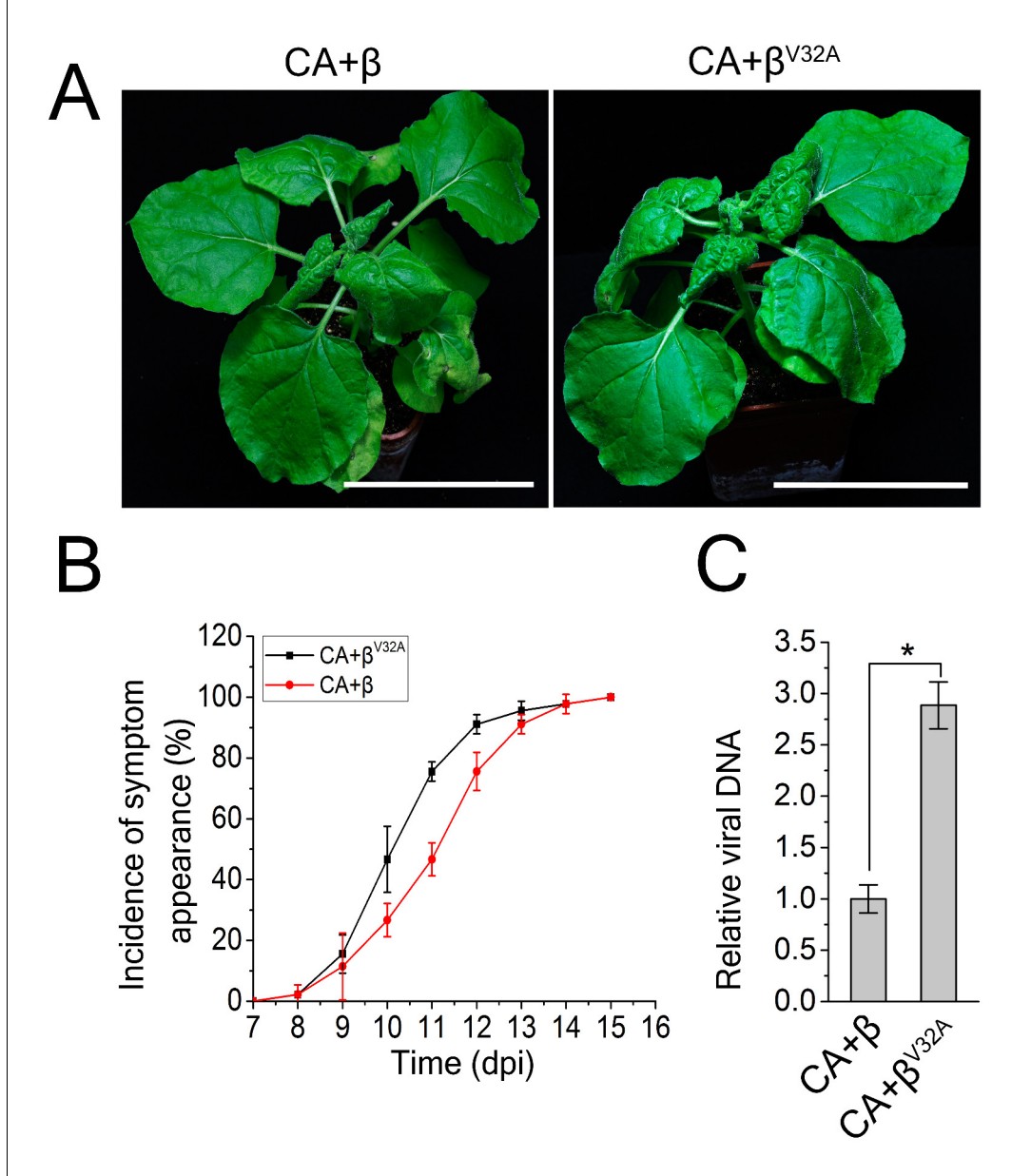

**Figure 4.** A V32A point mutation in $\beta$C1 enhanced CLCuMuV infection. (**A**) CLCuMuB mutant ($\beta^{V32A}$), which encodes a mutant $\beta$C1$^{V32A}$, caused the enhanced viral symptom compared to wild type CLCuMuB ($\beta$) when co-infected with CLCuMuV (CA). The pictures were taken at 12 dpi. A V32A point mutation in $\beta$C1 ($\beta$C1$^{V32A)}$ eliminates its interaction with NbATG8f. (**B**) The incidence of symptom appearance at different time points of post infection. Symptom was indicated as the appearance of curled leaf caused by the infection with CA+$\beta$ or CA+$\beta^{V32A}$. (**C**) Relative viral accumulation of CLCuMuV DNA. Real-time PCR analysis of *V1* gene from CLCuMuV was used to determine viral DNA level. Values represent means ± SE from three independent experiments. (*) $p<0.05$.

*ATG7*- silenced plants. As expected, we observed YFP-$\beta$C1 in the vacuoles in the non-silenced control plants. However, YFP-$\beta$C1 accumulated mostly in cytoplasm in *ATG5*- and *ATG7*- silenced plants (***Figure 5A***). Similarly, YFP-$\beta$C1$^{V32A}$ also accumulated mostly in cytoplasm in all plants, regardless of whether *ATG5 / ATG7* was or not silenced (***Figure 5—figure supplement 1***). In addition, silencing of either *ATG5* or *ATG7* resulted in more accumulation of YFP-$\beta$C1 but not YFP-$\beta$C1$^{V32A}$ (***Figure 5B*** and ***Figure 5—figure supplement 1***).

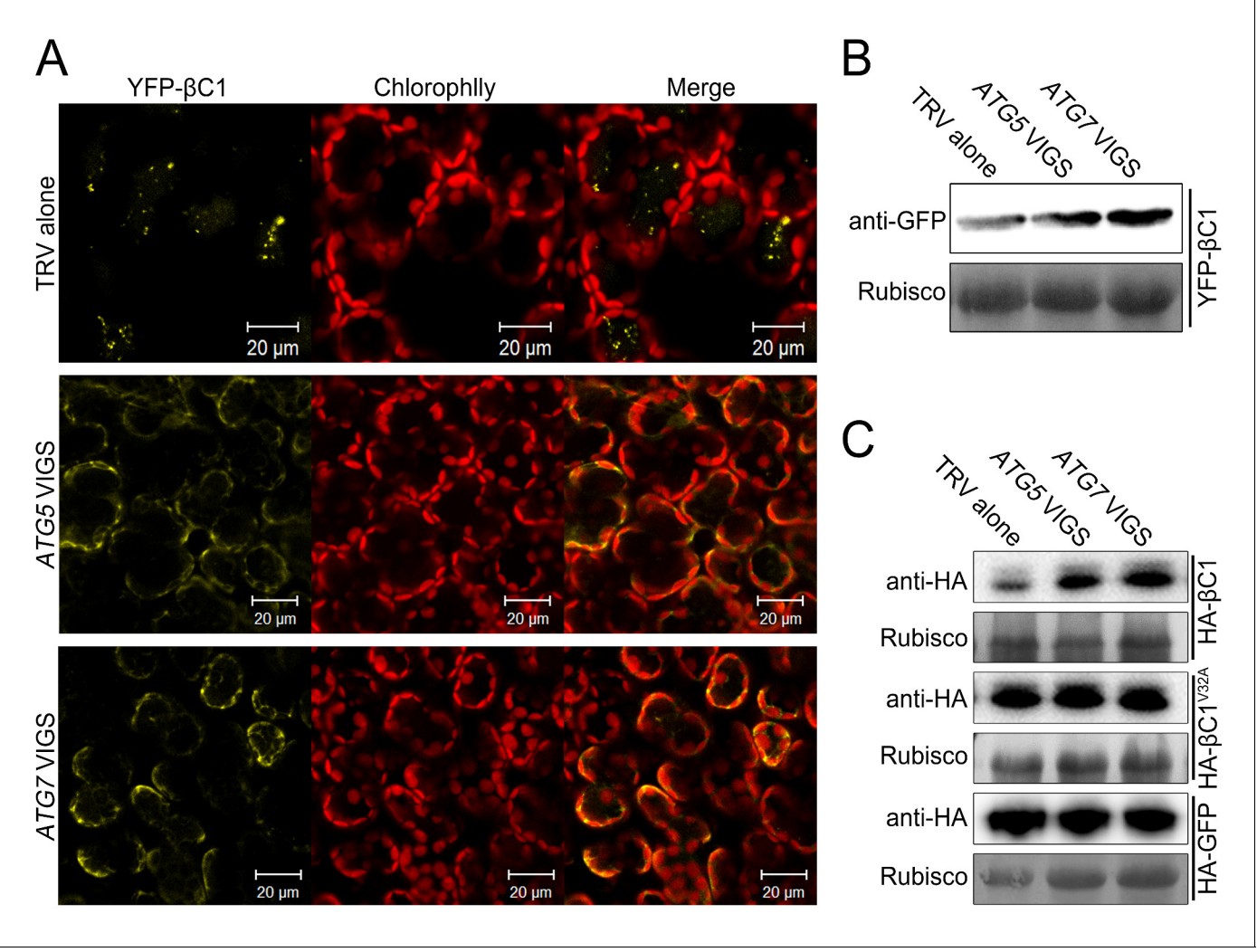

**Figure 5.** βC1 proteins is targeted for autophagic degradation. (A) Confocal microscopy images of YFP-βC1 in mesophyll cells of N. benthamiana leaves. YFP-βC1 was transiently expressed in non-silenced control (TRV alone), *ATG5* or *ATG7* silenced plants. The confocal microscope images of mesophyll cells were taken at 60 hpi. (B) Western blot analyses of YFP-βC1 construct from the same experiments as in (A). Level of the fusion protein, YFP-βC1, was detected with anti-GFP polyclonal antibody. (C) Silencing of either *ATG5* or *ATG7* enhanced the accumulation of HA-βC1, but not HA-βC1V32A. Each expression constructs were agroinfiltrated into *N. benthamiana* leaf. At 60 hpi leaf lysates were separated by SDS-PAGE and fusion proteins were detected by anti-GFP or anti-HA antibodies.

The following figure supplements are available for figure 5:

**Figure supplement 1.** Silencing of either *ATG5* or *ATG7* has no effect on localization of YFP-βC1V32A.

**Figure supplement 2.** Silencing of either *ATG5* or *ATG7* has no effect on transcript levels of target genes.

We also tested the impact of autophagy on HA-βC1. Silencing of either *ATG5* or *ATG7* increased the accumulation of βC1 but did not affect the level of βC1V32A mutant protein or GFP alone (in the control; *Figure 5C*). However, no difference in the RNA level of the βC1 construct was detected in *NbATG5*- or *NbATG7*-silenced plants (*Figure 5—figure supplement 2*).

These results strongly suggest that βC1 is targeted by autophagy for the degradation, and the interaction of βC1 with NbATG8 is required for the autophagic degradation of βC1.

## Autophagy functions as an antiviral mechanism against other geminiviruses

We then investigated the effects of autophagy on *Tomato yellow leaf curl virus* (TYLCV) and *Tomato yellow leaf curl China virus* (TYLCCNV). Silencing of *ATG5* and *ATG7* caused more severe disease symptoms and enhanced viral DNA accumulation compared to the control, and silencing of *GAPCs* reduced the severity of viral disease symptoms and viral DNA levels in plants infected by TYLCV or TYLCCNV (*Figure 6*). These results suggest that autophagy may have evolved as a general antiviral mechanism against various geminiviruses.

## Joka2-mediated selective autophagy is not involved in antiviral defense against CLCuMuV infection

Joka2/NBR1 is the sole known selective autophagy cargo receptor in plants. To investigate the potential role of selective autophagy in antiviral defense against CLCuMuV infection, we silenced *Joka2* in *N. benthamiana* using TRV-based VIGS (*Figure 7A*). *Joka2* mRNA levels were significantly reduced in *Joka2*-silenced plants compared to non-silenced control plants (*Figure 7B*). However, silencing of *Joka2* had no effect on CLCuMuV infection of plants with CA+$\beta$ (*Figure 7C*). These results suggest that *Joka2*-mediated selective autophagy is not involved in antiviral defense against CLCuMuV infection.

## Discussion

Autophagy is known to play an important role in disease resistance or susceptibility to various pathogens in plants (*Han et al., 2011*). However, how autophagy is linked to plant immunity remains unknown. In this study, we show that geminivirus CLCuMuV infection activates autophagy and that autophagy targets the virulence protein $\beta$C1 for degradation. Further, we demonstrated for the first time that autophagy plays an active role as an antiviral mechanism in compatible plant-virus interactions.

Autophagy acts as a defense mechanism against some invading intracellular pathogens in mammalian systems (*Boyle and Randow, 2013*; *Randow and Youle, 2014*; *Paul and Münz, 2016*). Plant cells also employ autophagy to defend themselves against several pathogens (*Han et al., 2011*; *Li et al., 2016*). Autophagy positively regulates plant resistance against necrotrophic pathogens (*Lai et al., 2011*; *Lenz et al., 2011*; *Kabbage et al., 2013*) but negatively affects plant resistance against the biotrophic pathogen powdery mildew (*Wang et al., 2011*). Furthermore, silencing of *Joka2*, encoding a selective autophagy cargo receptor for polyubiquitinated cargoes, enhances susceptibility to *Phytophthora infestans* although it is not reported whether disrupting autophagy has an effect on plant response to this pathogen (*Dagdas et al., 2016*). However, we found that *Joka2* silencing had no effect on CLCuMuV infection (*Figure 7*), implying that Joka2-mediated selective autophagy is not involved in antiviral defense against CLCuMuV in plants. CLCuMuB $\beta$C1 may not interfere with Joka2-mediated selective autophagy by depleting Joka2 out of ATG8 complexes because it has a non-classic ATG8-interacting motif (see below). In addition, silencing of autophagy-related genes caused cell death induced by *Tobacco mosaic virus* (TMV) to spread in inoculated leaves of *N. benthamiana* plants containing the TMV resistance gene *N* but had no effect on plant systemic resistance against TMV (*Liu et al., 2005*). However, the role of autophagy in compatible plant-virus interactions has been unclear. Here, we showed that disrupting autophagy enhanced plant susceptibility to three different DNA viruses, while activating autophagy enhanced plant resistance to viral infection (*Figure 3* and *Figure 6*), suggesting that autophagy plays an active antiviral role in compatible plant-virus interactions.

Autophagy can either facilitate or suppress viral infection in mammalian cells (*Orvedahl et al., 2010*; *Chiramel et al., 2013*; *Paul and Münz, 2016*; *Wang et al., 2016b*). However, little is known about whether and/or how autophagy affects viral infection in plants. Plant viruses encode antiviral RNA silencing suppressors known as VSRs. Based on the data from autophagy inhibitor 3-MA treatment, the VSR protein P0 from Polerovirus is thought to trigger the autophagic degradation of AGO1, a component of the cellular RNAi-based antiviral defense machinery (*Derrien et al., 2012*). Similarly, the VSR protein VPg from *Turnip mosaic virus* is also reported to mediate the degradation of the cellular RNAi-based antiviral defense component SGS3, partially via autophagy (*Cheng and*

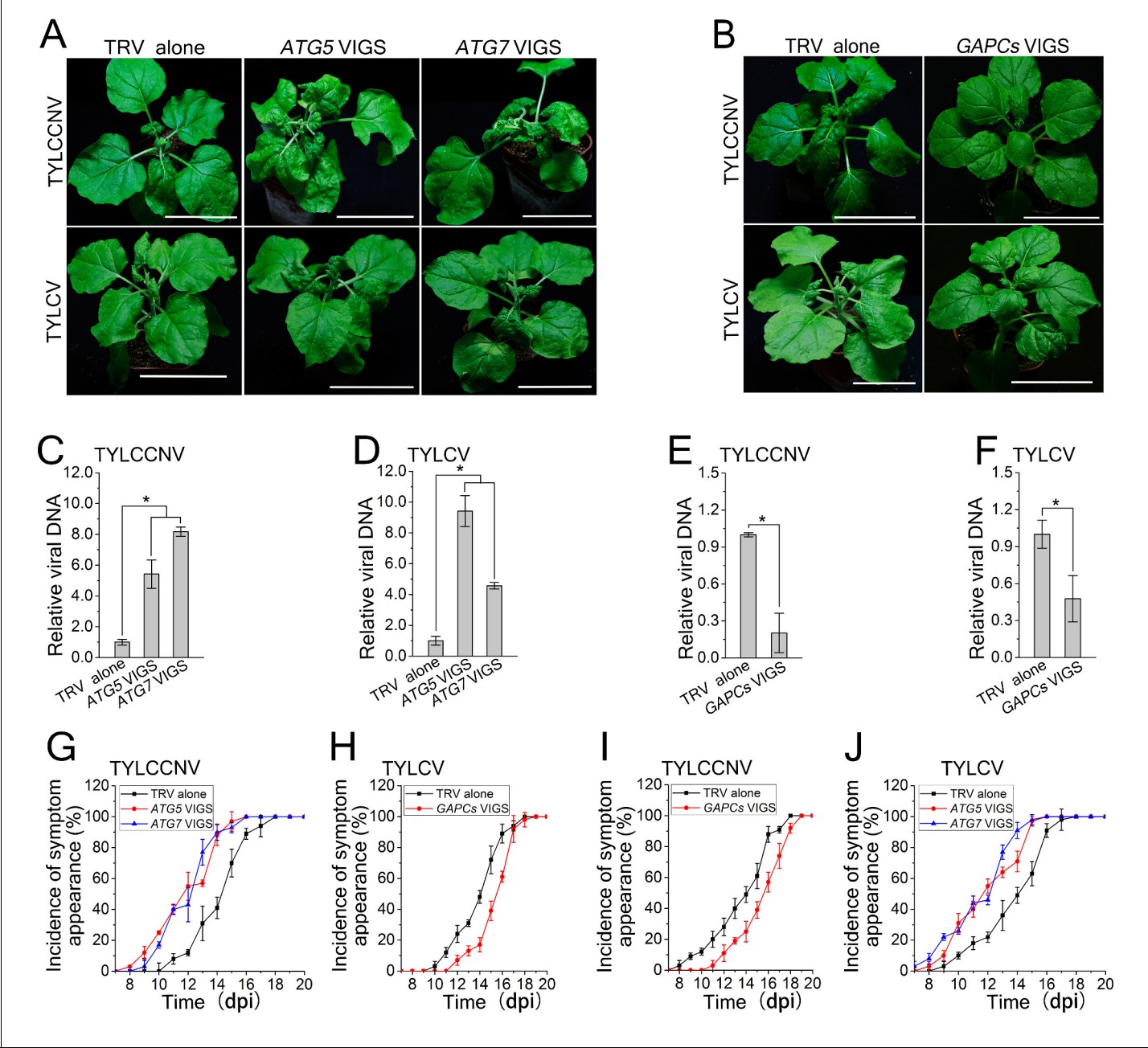

**Figure 6.** Autophagy regulates viral infection of TYLCV and TYLCCNV. (**A**) Viral symptoms in *ATG5*- and *ATG7*-silenced plants at 12 dpi. Bar represents 7 cm. (**B**) Viral symptoms in *GAPCs*–silenced (*GAPCs* VIGS) plants at 15 dpi. Bar represents 7 cm. (**C**) and (**D**) Relative viral DNA accumulation in *ATG5*- and *ATG7*-silenced plants. Real-time PCR analysis of *V1* gene from TYLCCNV (**C**) or TYLCV (**D**) was used to determine viral DNA level in infected or control (TRV alone) plants. Values represent means ± SE from three independent experiments. (*) $p < 0.05$. (**E**) and (**F**) Relative viral DNA accumulation in *GAPCs*–silenced plants. Real-time PCR analysis of *V1* gene from TYLCCNV (**E**) or TYLCV (**F**) was used to determine viral DNA level in infected or control (TRV alone) plants. Values represent means ± SE from three independent experiments. (*) $p < 0.05$. (**G**), (**H**), (**I**) and (**J**). The incidence of symptom appearance at different time points of post infection in VIGS plants. Symptom was indicated as the appearance of curled leaf caused by TYLCCNV in *ATG5*- and *ATG7*-silenced (**G**) and *GAPCs*–silenced (**I**) plants or TYLCV infection in *ATG5*- and *ATG7*-silenced (**H**) and *GAPCs*–silenced (**J**) plants. Values represent means ± SE from three independent experiments.

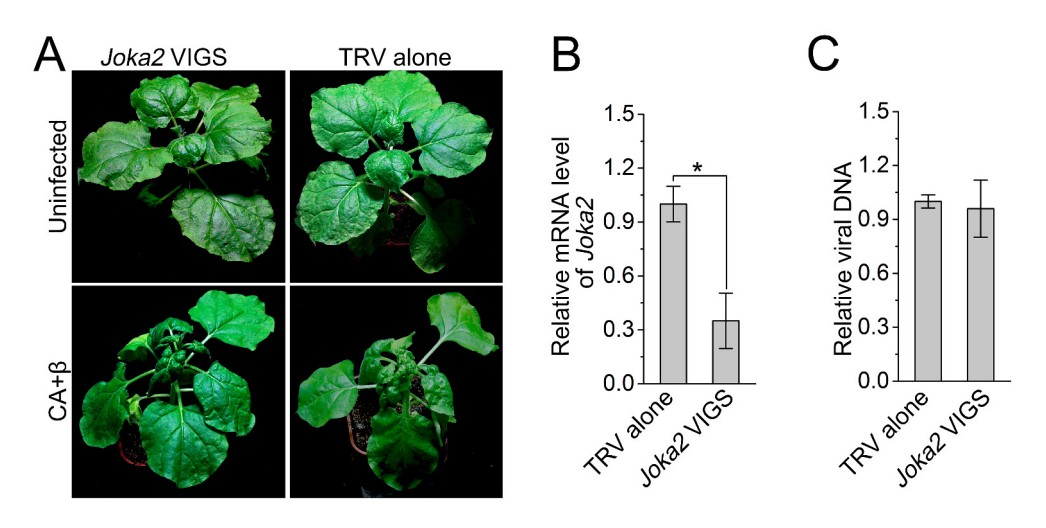

**Figure 7.** Silencing of *Joka2*, a plant selective autophagy cargo receptor, has no effect on CLCuMuV infection. (**A**) Viral symptoms in *Joka2*-silenced *N. benthamiana* at 12 dpi. (**B**) mRNA level of *Joka2* was reduced in *Joka2*-silenced plants. Real-time RT-PCR of *Joka2* was used to determine mRNA level. Values represent means ± SE from three independent experiments. (*) p<0.05. (**C**) Relative viral DNA accumulation in *Joka2*–silenced plants. Real-time PCR analysis of CLCuMuV *V1* gene was used to determine viral DNA level. Values represent means ±SE from three independent experiments. (*) p<0.05.

*Wang, 2016*). Based on the data from an autophagy inhibitor treatment and silencing of an autophagy-nonspecific ATG gene *Beclin 1*, 2b protein from *Cucumber mosaic virus* is thought to be targeted for degradation by autophagy through the calmodulin-like protein rgsCaM (*Nakahara et al., 2012*). In addition, the in vitro application of an autophagy inhibitor wortmannin partially inhibited proteolysis of TYLCV proteins (*Gorovits et al., 2016*). In our study, by silencing of two ATG genes (*ATG5* and *ATG7*), we clearly show that CLCuMuB βC1 is degraded by autophagy. Further, autophagy genes are transcriptionally up-regulated by infection by some geminiviruses in plants (*Ascencio-Ibáñez et al., 2008*; *Miozzi et al., 2014*). These observations indirectly suggest that autophagy may contribute to plant immunity during compatible plant–virus interactions. In this study, we showed that disrupting autophagy reduces plant resistance against three different plant viruses, whereas activating autophagy enhances this resistance. To the best of our knowledge, in this study, we provide the first direct evidence that autophagy functions as an antiviral mechanism in compatible plant-virus interactions.

CLCuMuB βC1 is predicted to contain two ATG8-interacting motifs (AIMs) (*Kalvari et al., 2014*). Interestingly, we found that an approximately 11-amino-acid motif (LVSTKSPSLIK) comprising residues 30 to 40 of βC1, but not the predicted AIMs, is responsible for the interaction of βC1 with NbATG8f. Furthermore, a point mutation at position 32 from valine to alanine in this motif eliminated the βC1-NbATG8 interaction. In addition, we found that βC1 interacted with at least four NbATG8 homologs. ATG8 proteins interact with some cargo receptors, leading to autophagic degradation of the cargoes (*Johansen and Lamark, 2011*). We found that disrupting autophagy increased the accumulation of βC1 protein but had no effect on the accumulation of βC1$^{V32A}$ (*Figure 5*), suggesting that βC1 is targeted for autophagic degradation. Furthermore, YFP–βC1 co-localized with CFP-NbAt8f-positive autophagic bodies in vacuoles of leaf mesophyll cells (*Figure 1—figure supplement 3* and *Video 1*), suggesting that βC1 is targeted for autophagic degradation by binding to ATG8s.

In this study, disrupting autophagy reduced plant resistance against TYLCV, while activating autophagy enhanced plant resistance against this virus. Interestingly, autophagy also participates in resistance to TYLCV in whiteflies (*Wang et al., 2016a*). TYLCV induces protein aggregation in plants and whiteflies, and viral proteins (mostly viral coat protein) and ATG8 co-exist in these TYLCV-induced aggregates (*Gorovits et al., 2016*). Further, some AIMs are evolutionarily conserved among

plant and animal proteins (*Kalvari et al., 2014*; *Dagdas et al., 2016*), it is possible that autophagy uses similar mechanism to degrade TYLCV protein(s) in plants and whiteflies.

The $\beta$C1 protein from TYLCCNV $\beta$-satellite interacts with and is phosphorylated by SnRK1 (sucrose-nonfermenting1-related kinase 1), a plant ortholog of budding yeast SNF1 and mammalian AMPK (AMP-activated protein kinase) (*Shen et al., 2011*). AMPK promotes autophagy by directly phosphorylating different protein substrates involved in the initiation phase of autophagy (*Cardaci et al., 2012*). Interestingly, TYLCCNB $\beta$C1 also contains the polypeptide sequence LASTK-SPALAK at residues 30–40, which is similar to the ATG8-binding motif of CLCuMuB $\beta$C1 (LVSTKSPSLIK). Moreover, a mutation in this motif affects TYLCCNV infection (*Shen et al., 2011*). It is possible that TYLCCNB $\beta$C1 is also targeted for autophagic degradation. Consistent with this hypothesis, disrupting autophagy reduced plant resistance against TYLCCNV, while activating autophagy enhanced plant resistance against this virus (*Figure 6*).

Disrupting autophagy increased viral DNA accumulation, while enhancing autophagy inhibited this process (*Figure 3*). Importantly, the elimination of the interaction of $\beta$C1 with NbATG8f resulted in enhanced viral infection, suggesting that the $\beta$C1-NbATG8 interaction is essential for the autophagy-mediated antiviral defense response against CLCuMuV. Thus, we provide compelling evidence that autophagy represents a novel antiviral strategy that involves targeting viral proteins for degradation and inhibiting viral infection in plants. This unexpected discovery may facilitate the development of new strategies to protect plants from viral invasion.

In summary, we provide direct evidence that autophagy functions as a novel antiviral mechanism in plants. Interestingly, CLCuMuB $\beta$C1 is an ATG8-binding protein, as well as a strong silencing suppressor. This finding suggests that a delicate balance between viral pathogenesis and different host antiviral immunity mechanisms, such as autophagy and RNA silencing, has developed during plant–virus co-evolution. Plants may have evolved to suppress viral infection by targeting viral protein(s) for autophagic degradation. Indeed, we showed that disrupting autophagy enhanced plant susceptibility to three different viruses, whereas increasing autophagy enhanced plant resistance against these viruses. On the other hand, possessing very strong virulence may not be the best strategy for the long-term survival of viruses in the battle between plants and viruses. In this scenario, plant viruses may have evolved the ability to use the host's cellular autophagy pathway to reduce their virulence through partial autophagic degradation of some viral virulence factors such as $\beta$C1. Thus, the virus would not completely destroy the plant cell or totally evade other host defense mechanisms such as RNA silencing or DNA methylation. Consistent with this idea, plant viruses can establish latent, mild, or severe infection in plants, but they rarely kill their hosts.

## Materials and methods

### Plant materials

*GFP*-transgenic16c line or wild type *N. benthamiana* plants were grown in pots placed in growthrooms at 25°C under a 16-h-light/8-h-dark cycle.

### Plasmid constructs

Vectors pTRV1 (*Liu et al., 2002*) and pTRV2-LIC (*Dong et al., 2007*) were described previously. pTRV2-Nb*ATG5* and pTRV2-Nb*ATG7* were described (*Wang et al., 2013*). VIGS construct of Nb*GAPCs* was described (*Han et al., 2015*). pTRV2-*NbJoka2* was generated by cloning *NbJoka2* cDNA fragment into pTRV2-LIC. Approximately 200 bp of mGFP5-ER were cloned into pTRV2 by specific primers.

The full-length infectious clones of CLCuMuV (GQ924756.1) and CLCuMuB (GQ906588.1) were described (*Jia et al., 2016*). The full-length infectious clones of TYLCCNV and TYLCV were described (*Tao and Zhou, 2004*; *Zhang et al., 2009*).

$\beta$C1 and its mutant $\beta$C1$^{V32A}$ were cloned into pGEX4T-1 vector to express GST-tagged fusion proteins in *Escherichia coli*. Full-length cDNA of Nb*ATG8f* was cloned into pET28a to express Nb*ATG8f*-6×His in *E. coli*.

CFP-Nb*ATG8f*, GFP-Nb*ATG8f*, GFP-Nb*ATG8c*, GFP-Nb*ATG8d*, GFP-Nb*ATG8i*, cYFP-Nb*ATG8f*, nYFP-$\beta$C1, GFP-$\beta$C1, YFP-$\beta$C1, HA-$\beta$C1, HA-nLUC, nYFP-$\beta$C1$^{V32A}$, GFP-$\beta$C1$^{V32A}$, YFP-$\beta$C1$^{V32A}$, HA-$\beta$C1$^{V32A}$, and HA-nLUC were obtained respectively by overlapping PCR, and then cloned between

the duplicated CaMV 35S promoter and the NOS terminator of pJG045, a pCAMBIA1300-based T-DNA vector (*Du et al., 2013*).

## Co immunoprecipitation (Co-IP) assay

For co-IP assays, total proteins from *N. benthamiana* leaves (1 g leaf tissues for each sample) were extracted in ice-cold immunoprecipitation buffer (10% [v/v] glycerol, 25 mM Tris, pH 7.5, 150 mM NaCl, 1× protease inhibitor cocktail [Roche], and 0.15% [v/v] Nonidet P-40) as described (*Wang et al., 2015*). Protein extracts were incubated with GFP-Trap beads (ChromoTek) for 3 hr at 4°C. The precipitations were washed four times with ice-cold immunoprecipitation buffer at 4°C and were analyzed by immunoblot using anti-HA (Cell Signaling Technology), or anti-GFP (ChromoTek) antibodies.

## VIGS assay

TRV-mediated VIGS assays were performed as described (*Liu et al., 2005*).

## Confocal microscopy and TEM

Confocal imaging was performed as described (*Han et al., 2015*). The leaves were agroinfiltrated with autophagy marker CFP-NbATG8f for 60 hr expression, followed by an additional infiltration with 20 uM E-64d for 8 hr before being monitored by a Zeiss LSM 710 three-channel microscope with an excitation light of 405 nm, and the emission was captured at 454 to 581 nm. TEM observation was performed as described (*Wang et al., 2013*).

## GST pull-down assays

GST pull-down assays were performed as described previously (*Zhao et al., 2013*). GST-$\beta$C1 and NbATG8f-6×His fusion proteins were produced in BL21 (DE3) cells (Stratagene).

## Yeast two-hybrid screen and interaction assays

For yeast two-hybrid interaction assay, CLCuMuB *$\beta$C1* was PCR amplified and cloned into yeast vector pYL302 to generate the LexA DNA binding domain (BD) containing bait vectors. *NbATG8f* was PCR amplified and cloned into the B42 activation domain (AD)-containing vector pSAH20b. The interaction assay and yeast two-hybrid screen was performed as described (*Du et al., 2013*).

## Real-time PCR analysis

For expression analysis of *NbATG* genes, real-time RT-PCR was performed as described (*Wang et al., 2013*). *eIF4a* was used as the internal control.

For quantification of viral DNA loads, two DNA standard curves were generated. Single copy of target viral genome DNA was cloned into pMD19-T (TaKaRa, Japan) and was used as standard viral DNA (SVD) while the genome DNA of healthy *N. benthamiana* was served as standard genome DNA (SGD). Standard curves were generated from ten-fold dilutions of both SVD and SGD. Approximately 200 bp fragment of *V1* gene from CLCuMuV/TYLCY/TYLCCNV and 61 bp region of *eIF4$\alpha$* gene were amplified by employing SYBR green based real-time PCR to produce standard curves. Viral DNA load in infectious plants was determined according to standard curves of SVD and SGD. Results were expressed as fold change of virus DNA from virus infected plant tissue.

For quantification of TRV viral loads, real-time RT-PCR was performed with TRV *CP* - specific primers (*Zhao et al., 2013*). *eIF4a* was used as the internal control.

## Western blotting

For protein analysis, total proteins were extracted from *N. benthamiana* leaves using 2×Laemmli buffer. After boiling for 10 min, protein extracts were separated by SDS-PAGE for immunoblot analysis with indicated antibodies as described (*Wang et al., 2013*).

## BiFC assay

Citrine Yellow Fluorescent protein (YFP)-based Bimolecular Fluorescence Complementation (BiFC) assay was performed as described (*Burch-Smith et al., 2007*; *Jia et al., 2016*).

## Acknowledgements

We thank Professor Xueping Zhou for kindly providing the infectious clones of TYLCCNV and TYLCV. This work was supported by the National Basic Research Program of China (2014CB138400), the National Natural Science Foundation of China (31530059, 31421001, 31470254, and 31370180).

## Additional information

### Funding

| Funder | Grant reference number | Author |
|---|---|---|
| National Natural Science Foundation of China | 31421001 | Yijun Qi |
| National Natural Science Foundation of China | 31370180 | Yiguo Hong |
| National Natural Science Foundation of China | 31530059 | Yule Liu |
| National Natural Science Foundation of China | 31470254 | Yule Liu |
| National Basic Research Program of China | 2014CB138400 | Yule Liu |

The funders had no role in study design, data collection and interpretation, or the decision to submit the work for publication.

### Author contributions

YHa, Formal analysis, Validation, Investigation, Methodology, Writing—original draft, Writing—review and editing; AI, Formal analysis, Validation, Investigation, Writing—review and editing; QJ, NL, SH, Investigation; YWan, XZ, TC, LQ, YWan, JC, Investigation, Methodology; YQ, YHo, Funding acquisition, Writing—review and editing; YL, Conceptualization, Resources, Formal analysis, Supervision, Funding acquisition, Writing—original draft, Project administration, Writing—review and editing

### Author ORCIDs

Yakupjan Haxim, http://orcid.org/0000-0001-8559-0238
Yule Liu, http://orcid.org/0000-0002-4423-6045

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
