## [Decision Letter]

Thank you for submitting your article "Autophagy functions as an antiviral mechanism in plants" for consideration by *eLife*. Your article has been favorably evaluated by Detlef Weigel (Senior Editor) and three reviewers, one of whom, Jian-Min Zhou (Reviewer #1), is a member of our Board of Reviewing Editors.

The reviewers have discussed the reviews with one another and the Reviewing Editor has drafted this decision to help you prepare a revised submission.

Summary:

Autophagy is known to play an important role in disease resistance or susceptibility to various pathogens. Recent evidence also suggests that autophagy is targeted by a Phytophthora effector for virulence. However, how autophagy contributes to disease resistance remains unknown. In this study the authors show that geminivirus CLCuMuV infection activates autophagy and that autophagy targets the virulence protein βC1 for degradation in the vacuoles through an interaction between ATG8 and a specific motif in βC1. A mutation in this motif not only abolishes protein-protein interaction and βC1 degradation, but also enhances disease symptoms and viral DNA accumulation. Thus the study elegantly illustrate a mechanism by which autophagy directly destruct the virulence protein to confer resistance. Although a potential involvement of autophagy in viral protein degradation has been implicated by previous pharmacological studies, the current study represents the first demonstration of such a process and elucidates how this occurs. Overall the data are convincing and manuscript well-written. The reviewers raised several issues that require editorial changes:

Essential revisions:

1) The authors state "whether plants exploit autophagy to combat virus infection remains unclear"(Abstract) "However, the role of autophagy in plant immunity against intracellular plant pathogens such as viruses has remained unclear" (Discussion). These statements are not accurate. A role of autophagy in antiviral HR has been shown by the author (Cell 2005), and a role of autophagy in degrading viral proteins has been implicated by pharmacological studies. What is not known is the mechanism by which autophagy controls disease resistance. Whether this is through regulating immune signaling or direct destruction of virulence proteins remains to be shown. The authors should clearly describe previous and the current contribution.

2) Introduction, third paragraph: "a fungal effector…" should be an oomycete effector.

3) The title should be concise and be clear this is a mechanism against geminiviruses. There is no evidence the stated mechanism also applies to other viruses.

4) Several important references concerning the role of autophagy in viral infection in mammalian cells should be included to strengthen the Discussion (Chiramel et al., 2013 in Cells; Paul and Munc, 2016 in Adv Virus Res; Wang et al., 2016 in J Virol).

5) The work by Wang et al., 2016 "The autophagy pathway participates in resistance to tomato yellow leaf curl virus infection in whiteflies" should be cited and discussed.

6) In the Results section, the conclusion (subsection “βC1 is targeted by autophagy for its degradation”, last paragraph) is premature; let it be at the end of the subsection “The βC1-ATG8f protein interaction is involved in antiviral defense against CLCuMuV infection”.

7) Considering there are many ATG8 isoforms, does βC1 interact specifically with NbATG8f? Please discuss.

8) Why *Joka2* has no role should be discussed considering it is a major selective autophagy receptor in plants.

---

## [Author Response]

Essential revisions:

1) The authors state "whether plants exploit autophagy to combat virus infection remains unclear"(Abstract) "However, the role of autophagy in plant immunity against intracellular plant pathogens such as viruses has remained unclear" (Discussion). These statements are not accurate. A role of autophagy in antiviral HR has been shown by the author (Cell 2005), and a role of autophagy in degrading viral proteins has been implicated by pharmacological studies. What is not known is the mechanism by which autophagy controls disease resistance. Whether this is through regulating immune signaling or direct destruction of virulence proteins remains to be shown. The authors should clearly describe previous and the current contribution.

Thanks editors for these constructive suggestions. We have followed your suggestions and revised the manuscript.

2) Introduction, third paragraph: "a fungal effector…" should be an oomycete effector.

Corrected.

3) The title should be concise and be clear this is a mechanism against geminiviruses. There is no evidence the stated mechanism also applies to other viruses.

We have followed your suggestions.

4) Several important references concerning the role of autophagy in viral infection in mammalian cells should be included to strengthen the Discussion (Chiramel et al., 2013 in Cells; Paul and Munc, 2016 in Adv Virus Res; Wang et al., 2016 in J Virol).

We have included these references.

5) The work by Wang et al., 2016 "The autophagy pathway participates in resistance to tomato yellow leaf curl virus infection in whiteflies" should be cited and discussed.

We have cited this reference, and discussed this in the Discussion.

6) In the Results section, the conclusion (subsection “βC1 is targeted by autophagy for its degradation”, last paragraph) is premature; let it be at the end of the subsection “The βC1-ATG8f protein interaction is involved in antiviral defense against CLCuMuV infection”.

We have followed this suggestion.

7) Considering there are many ATG8 isoforms, does βC1 interact specifically with NbATG8f? Please discuss.

βC1 interacts at least four NbATG8 homologs. We have included this data in the revised manuscript, and discussed this. See Figure 1—figure supplement 2.

8) Why Joka2 has no role should be discussed considering it is a major selective autophagy receptor in plants.

We have discussed this issue in the revised manuscript.